# Serological Evidence of Severe Fever with Thrombocytopenia Syndrome Virus and IgM Positivity Were Identified in Healthy Residents in Vietnam

**DOI:** 10.3390/v14102280

**Published:** 2022-10-17

**Authors:** Xuan Chuong Tran, Sung Hye Kim, Jeong-Eun Lee, So-Hee Kim, Su Yeon Kang, Nguyen D. Binh, Pham V. Duc, Phan T. K. Phuong, Nguyen T. P. Thao, Wonwoo Lee, Joon-Yong Bae, Man-Seong Park, Misun Kim, Jeong Rae Yoo, Sang Taek Heo, Kyeong Ho An, Jung Mogg Kim, Nam-Hyuk Cho, Sun-Ho Kee, Keun Hwa Lee

**Affiliations:** 1Department of Infectious Diseases, Hue University of Medicine and Pharmacy, Hue 530000, Vietnam; 2Department of Microbiology and Environmental Biology & Medical Parasitology, Hanyang University College of Medicine, Seoul 04763, Korea; 3Department of Microbiology, Korea University College of Medicine, Seoul 02841, Korea; 4Department of Microbiology and Immunology, Seoul National University College of Medicine, Seoul 03080, Korea; 5Department of Internal Medicine, Jeju National University College of Medicine, Jeju 64231, Korea; 6Department of Microbiology, Graduate School of Dongguk University, Seoul 04620, Korea

**Keywords:** severe fever with thrombocytopenia syndrome virus (SFTSV), serological evidence, IgM positivity, healthy residents, Vietnam

## Abstract

Severe fever with thrombocytopenia syndrome (SFTS), an emerging tick-borne viral disease, is prevalent in East Asia and has also been reported in Southeast Asia since 2019. SFTS patients in Vietnam were first reported in 2019. However, the seroprevalence of severe fever with thrombocytopenia syndrome virus (SFTSV) in Vietnam has not been reported. To investigate the seroprevalence of SFTSV in Vietnam, we collected serum samples from 714 healthy residents in Thua Thien Hue and Quang Nam Province, Vietnam, and the seroprevalence of SFTSV was assessed using immunofluorescence antibody assay (IFA), Enzyme-Linked Immunosorbent Assays (ELISAs) and the 50% focus reduction neutralization test (FRNT50) assay. The seroprevalence of anti-SFTSV IgM or IgG was observed to be 3.64% (26/714), high IgM positivity was >80 (0.28%, 2/714) and the titer of neutralizing antibodies against SFTSV ranged from 15.5 to 55.9. In Pakistan, SFTSV infection confirmed using a microneutralization test (MNT) assay (prevalence is 2.5%) and ELISAs showed a high seroprevalence (46.7%) of SFTSV. Hence, the seroprevalence rate in Vietnam is similar to that in Pakistan and the number of SFTS patients could increase in Vietnam.

## 1. Introduction

Severe fever with thrombocytopenia syndrome (SFTS) is an emerging tick-borne disease caused by the *Dabie bandavirus* of the genus *Bandavirus* (formerly SFTS virus, SFTSV) [1,2,3]. Severe fever with thrombocytopenia syndrome virus (SFTSV) is an enveloped and tri-segmented (large (L), middle (M), and small (S)) negative-strand RNA virus and was first reported in rural areas of Hubei and Henan provinces in central China in 2009 [1,4]. 

The major clinical symptoms of severe fever with thrombocytopenia syndrome (SFTS) are acute and high fever (temperatures of 38 °C or more), thrombocytopenia (platelet count < 100,000/mm^3^), leucopenia, elevated levels of serum hepatic enzymes, gastrointestinal symptoms and multiorgan failure, with a 16.2 to 30% mortality rate, and effective antiviral therapy for SFTS virus (SFTSV) has not been available [1,4,5,6].

Atypical signs and symptoms as well as asymptomatic SFTSV infections have also been identified in patients and healthy residents who lived in endemic areas [2,3].

SFTS was first reported in China in 2009, South Korea in 2010 and Japan in 2013. SFTSV infection has also been first reported in Southeast Asia since 2019 (Vietnam in 2019, Myanmar, Taiwan, Thailand and Pakistan in 2020) [1,7,8,9,10,11,12,13].

Although most SFTSV infections occur through bites by the ticks *Hemaphysalis longicornis, Amblyomma testudinarium* and *Ixodes nipponensis*, transmission can also occur through close contact with an infected patient or animal [1,2,3,14].

As cases of SFTS have been reported in Vietnam, a study of the seroprevalence of SFTSV infection is important for determining the risk of infection in this country [9]. Therefore, we investigated SFTSV seroprevalence among healthy residents in Vietnam to understand the disease burden.

## 2. Materials and Methods 

To investigate the seroprevalence of SFTSV in Vietnam, we collected serum samples from 714 healthy residents in Thua Thien Hue and Quang Nam Province, Vietnam, encompassing areas of large forests, hilly landscapes and grasslands, from 1 October 2017 to 30 September 2018. Most of the 714 residents were farmers, with a tick exposure risk. Informed consent was obtained from all participants, the study is in accordance with relevant guidelines and regulations and this was approved by the Institutional Review Board (IRB) of Hue University Hospital, Hue, Vietnam.

### 2.1. Measurement of Anti-SFTSV IgG and IgM Using Immunofluorescence Antibody Assay (IFA)

Serological testing for the presence of anti-SFTSV IgM and IgG was performed using IFA assay as previously described [2,6]. 

For IFA, Vero E6 cells infected with SFTSV were incubated in a 5% CO_2_ incubator at 37 °C for five days. The cells were harvested, inoculated onto Teflon-coated well slides and fixed with acetone. IFA was carried out using a patient’s serum as the primary antibody and fluorescein-labeled anti-human IgG or IgM secondary antibodies (Thermo Fisher Scientific, Waltham, MA, USA). Serum specimens were diluted twofold from 10 to 2560. The incubation time of a patient’s serum was extended to 90 min for IgM detection, whereas a 30-min incubation was performed for IgG detection. A monoclonal anti-SFTSV N antibody (produced in our laboratory) was used as the positive control.

### 2.2. Measurement of Anti-SFTSV IgG and IgM Using Enzyme-Linked Immunosorbent Assays (ELISAs)

ELISAs for anti-SFTSV IgM and IgG in 26 serum samples that were positive by IFA (IgM or IgG) were performed as previously described [6], Bore Da Biotech Co., Ltd., Gyeonggi-do, Korea, http://www.boreda.com (accessed on 15 September 2022), with SFTSV nucleocapsid protein (NP) as the viral antigen. 

### 2.3. The 50% Focus Reduction Neutralization Test (FRNT50) Assay

To determine the titers of neutralizing antibody to SFTSV in human sera, we performed a FRNT50 assay, as previously described, on 10 serum samples that were strongly positive for IFA (IgM or IgG) (Table 1) [15]. 

To perform the FRNT50 assay, serum samples were heat-inactivated at 56 °C for 30 min and serially diluted 2-fold from 1:10 to 1:160. Each dilution was mixed and preincubated with an equal volume of solution containing SFTSV (100 focus-forming units (FFU)) at 4 °C for 1 h. The mixture was inoculated into Vero E6 cells in 24-well plates followed by incubation for 1.5 h at 37 °C. Culture medium was used as a control. After incubation, the cells were overlaid with 0.5 mL of Dulbecco’s modified Eagle’s medium (2× DMEM) containing 0.8% carboxymethyl cellulose and 2% FBS, and the cells were cultured for an additional 5 days. The cells were fixed with 100% methanol and incubated with 200 μL/well rabbit anti-SFTSV NP antibodies for 90 min at room temperature (RT), followed by incubation with AP-conjugated secondary antibodies. Visualization of SFTSV-infected cell foci was performed using NBT/BCIP tablets as the substrate. The plaque reduction percentage was calculated by using the formula ((number of SFTSV-infected cell foci diluted without serum) − (number of SFTSV-infected cell foci diluted with serum)) × 100/SFTSV-infected cell foci diluted without serum. FRNT50 titers were calculated based on this plaque reduction percentage using the (log (inhibitor) vs. normalized response) equation with GraphPad Prism 8.0.

### 2.4. Real-Time RT-PCR for Molecular Diagnosis

RNA was extracted from 26 serum samples that were positive by IFA (IgM or IgG) using a QIAamp Viral RNA Mini Kit (Qiagen, Hilden, Germany). Real-time RT-PCR of the partial small (S) segment of SFTSV was performed for molecular diagnosis [2].

## 3. Results 

### 3.1. Seropositive Rate of IgG and IgM Based on IFA and ELISA 

A total of 714 healthy residents were enrolled in this study. Serological testing for the presence of anti-SFTSV IgM and IgG was performed using IFA assay and IgM (24/714, 3.36%) and IgG (26/714, 3.64%) were detected in 714 serum samples from healthy residents; the IgM levels of ZC668 and ZH128 were high (>160 and 80, respectively) (Table 1 and Figure 1).

Twenty-six serum samples that were positive by IFA (IgM or IgG) were evaluated for IgM (30.8%, 8 in 26) and IgG (96.2%, 25 in 26) using ELISAs for anti-SFTSV IgM and IgG (Table 1).

These 26 healthy residents had no typical symptoms or other infectious diseases or contact with SFTS patients during the period of sample collection. 

### 3.2. The Titers of Neutralizing Antibody to SFTSV

We also performed a FRNT50 assay as previously described on 10 serum samples that were strongly positive for IFA (IgM or IgG) and the results of FRNT50 showed titers of neutralizing antibodies ranging from 15.5 to 55.9 (Table 1).

### 3.3. Molecular Diagnostic of SFTSV 

Real-time RT-PCR of the partial small (S) segment of SFTSV was performed for molecular diagnosis about 26 serum samples that were positive by IFA and ELISA (IgM or IgG) and the results of real-time RT-PCR were found to be negative [2].

## 4. Discussion

SFTSV infection is prevalent in East Asian countries (China, South Korea and Japan) and has been reported in Southeast Asian countries (Vietnam, Myanmar, Thailand, Taiwan and Pakistan) since 2019 [1,7,8,9,10,11,12,13].

The seroprevalence ranges from 0.23% to 9.17% in China, from 1.9% to 7.7% in South Korea and from 0.14% to 0.3% in Japan, and prevalence rate is 2.5% in Pakistan [13,16,17,18,19,20]. The IgG seroprevalence of SFTSV among healthy residents on Jeju Island, South Korea, is 2.4% and the titer of neutralizing antibodies against SFTSV ranges from 24.25 to 104.3 [15,21].

In this study, we examined the seroprevalence of anti-SFTSV IgM or IgG among healthy residents in Vietnam and found 3.64% positivity (26/714); the titer of neutralizing antibodies against SFTSV ranged from 15.5 to 55.9 (Table 1). 

Hence, the seroprevalence rate in Vietnam is similar to that in Pakistan and the titer of neutralizing antibodies against SFTSV ranges is lower than that in South Korea [13,20].

*A. testudinarium*, a vector of SFTSV, has been found in Vietnam, and virus-bearing *A. testudinarium* may play a role in circulating SFTSV in Vietnam, possibly contributing to the SFTSV seroprevalence rate in this country [9].

Asymptomatic SFTSV infections have also been identified in healthy people; IgM, SFTSV RNA were detected, and SFTSV were also isolated in healthy people in China and South Korea [2,3]. 

The anti-SFTSV IgM levels of ZC668 and ZH128 (>160 and 80, respectively) were high. Therefore, asymptomatic SFTSV infections have also been identified in healthy people in Vietnam and this result has implications for SFTS outbreak control in Vietnam.

In conclusion, we report the seroprevalence of SFTSV infection in healthy residents in Vietnam and our results show that the seroprevalence rate in Vietnam is similar to that in Pakistan, the titer of neutralizing antibodies against SFTSV ranges is lower than that in South Korea and asymptomatic SFTSV infections have been identified in healthy people in Vietnam, suggesting that the disease burden is similar to that in Pakistan [2,3,13,15].

Therefore, further ecological study of the virus in ticks and animals and epidemiological and clinical research are needed to better understand the epidemiology and transmission dynamics of SFTSV in Vietnam because the mortality rate of SFTSV infection is high and threatens public health in the country [9].

## Figures and Tables

**Figure 1 viruses-14-02280-f001:**
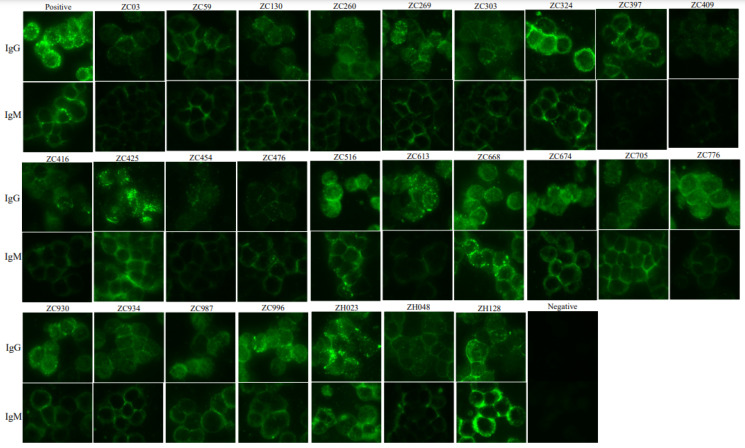
Immunofluorescence assay for IgG and IgM in healthy residents in Vietnam. Serum dilution: 1:80.

**Table 1 viruses-14-02280-t001:** Baseline characteristics of healthy residents in SFTSV IgM or G-positive group.

No.	Age (Year)/Sex	Occupation	Date of Sampling	Status	IFA (IgM)	ELISA (IgM)	IFA (IgG)	ELISA (IgG)	FRNT50 Titer *	RT-PCR for SFTSV	Study Site
ZC03	57/F	Small trader	25 August 2018	Healthy	P	N	60	P	N/A	N	Thua Thien Hue Province
ZC59	36/F	Farmer	25 August 2018	Healthy	P	P	10	P	N/A	N	Thua Thien Hue Province
ZC130	30/F	Farmer	25 August 2018	Healthy	P	N	10	P	N/A	N	Thua Thien Hue Province
ZC260	76/F	Retired	25 August 2018	Healthy	P	N	10	P	N/A	N	Thua Thien Hue Province
ZC269	16/F	Student	25 August 2018	Healthy	P	N	40	P	1:10 (15.8)	N	Thua Thien Hue Province
ZC303	64/M	Farmer	25 August 2018	Healthy	P	N	10	P	N/A	N	Thua Thien Hue Province
ZC324	55/F	Farmer	25 August 2018	Healthy	P	P	160	P	1:20 (29.6)	N	Thua Thien Hue Province
ZC397	46/M	Teacher	25 August 2018	Healthy	N	P	10	P	N/A	N	Thua Thien Hue Province
ZC409	50/M	Small trader	25 August 2018	Healthy	N	N	P	P	N/A	N	Thua Thien Hue Province
ZC416	25/M	Worker	25 August 2018	Healthy	P	N	10	P	N/A	N	Thua Thien Hue Province
ZC425	56/F	Farmer	25 August 2018	Healthy	10	N	2560	P	1:40 (49.0)	N	Thua Thien Hue Province
ZC454	23/M	Driver	25 August 2018	Healthy	P	N	10	P	N/A	N	Thua Thien Hue Province
ZC476	33/F	Farmer	25 August 2018	Healthy	P	P	10	P	N/A	N	Thua Thien Hue Province
ZC516	25/M	Farmer	8 September 2018	Healthy	40	N	P	P	1:20 (20.6)	N	Quang Nam Province
ZC613	63/M	Free worker	8 September 2018	Healthy	P	N	160	P	1:10 (20.6)	N	Quang Nam Province
ZC668	50/F	Free worker	8 September 2018	Healthy	160<	P	P	P	1:40 (55.9)	N	Quang Nam Province
ZC674	35/M	Farmer	8 September 2018	Healthy	P	P	P	P	N/A	N	Quang Nam Province
ZC705	38/F	Farmer	8 September 2018	Healthy	10	N	10	P	N/A	N	Quang Nam Province
ZC776	35/F	Worker	8 September 2018	Healthy	P	N	20	P	1:20 (31.6)	N	Quang Nam Province
ZC930	23/M	Student	8 September 2018	Healthy	P	N	10	P	N/A	N	Quang Nam Province
ZC934	39/M	Farmer	8 September 2018	Healthy	P	N	P	P	N/A	N	Quang Nam Province
ZC987	47/M	Farmer	8 September 2018	Healthy	P	N	P	N	N/A	N	Quang Nam Province
ZC996	54/M	Farmer	8 September 2018	Healthy	20	P	20	P	1:20 (33.7)	N	Quang Nam Province
ZH023	22/F	Farmer	14 June 2018	Healthy	P	P	160	P	1:20 (20.0)	N	Thua Thien Hue Province
ZH048	31/F	Small trader	14 June 2018	Healthy	P	N	10	P	N/A	N	Thua Thien Hue Province
ZH128	55/M	Farmer	25 August 2018	Healthy	80	N	80	P	1:20 (35.2)	N	Thua Thien Hue Province

* Calculated titer using regression fit; F, female; M, male; P, positive; N, negative; N/A, not applicable.

## Data Availability

The data presented in this study are available in this article.

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
