# Peer review of "Serological Evidence of Severe Fever with Thrombocytopenia Syndrome Virus and IgM Positivity Were Identified in Healthy Residents in Vietnam"

_viruses, 2022, doi:10.3390/v14102280_

Round 1

Reviewer 1 Report (Previous Reviewer 2)

The authors report measuring the prevalence of anti-SFTSV antibodies in healthy individuals to estimate the disease burden of SFTS in Vietnam.  It is a resubmitted manuscript but in my opinion the review is not yet sufficient. In particular, the indicated references are not sufficient. I request to implement them. In addition the authors mention the large disease burden of SFTS in Vietnam in terms of seroprevalence but the incidence of SFTS patients in Vietnam is low. This point is not clear. I request to better clarify  this point providing a possible explanation for the phenomenon.

Author Response

The authors report measuring the prevalence of anti-SFTSV antibodies in healthy individuals to estimate the disease burden of SFTS in Vietnam. It is a resubmitted manuscript but in my opinion the review is not yet sufficient.
In particular, the indicated references are not sufficient. I request to implement them. 

: Dear reviewer, 

Thank you very much for your commentary. 

Could you suggest reference (s)? When you suggest reference (s), we will put this (or theses). 

In addition the authors mention the large disease burden of SFTS in Vietnam in terms of seroprevalence but the incidence of SFTS patients in Vietnam is low. 
This point is not clear. I request to better clarify this point providing a possible explanation for the phenomenon.

: Dear reviewer, 

First of all, thank you very much for your commentary. 

Reference (Emerg. Infect. Dis. 2020, 26 (7),1513-1516) is a serology study of SFTSV in Pakistan and confirmed SFTSV infection in Pakistan. 

Authors of this paper described below in this article. 
“In Pakistan, which is not endemic region of SFTSV, the ELISA revealed a high seroprevalence (46.7%, 95% CI 44.3%-49.1%) of SFTSV and SFTSV infection were confirmed using MNT (microneutralization test), which revealed a low prevalence (2.5%, 95% CI 1.9%-3.4%)” (Ref. Serologic Evidence of Severe Fever with Thrombocytopenia Syndrome Virus and Related Viruses in Pakistan. Emerg. Infect. Dis. 2020, 26 (7),1513-1516).

In fact, Pakistan is not an endemic region of SFTSV. But, they described that “Our results indicate substantial activity of SFTSV and SFTSV-related viruses in this country” in this article. 

Therefore, we think that the seroprevalence rate in Vietnam is similar to that in Pakistan and put this in the manuscript.

Please see below. 
“Hence, the seroprevalence rate in Vietnam is similar to that in Pakistan and SFTS patients could be increased in the future in Vietnam” in line 35 and 36. 

Reviewer 2 Report (Previous Reviewer 1)

Tran XC et al. revised their manuscript again, and I reviewed the new version. They report measuring the prevalence of anti-SFTSV antibodies in healthy individuals in Vietnam with IFA, ELISA, and FRNT50 methods. It is a good report because such an epidemiological study in Vietnam has not been reported previously; howener, there are several points that need to be corrected and improved. 

Major comments

1.       About the section of Material and Methods. The authors should describe the cut-off level (value) for considering positive in the assay of IFA and ELISA. It is believed that the authors used the experimental system of Reference 4. According to Reference 4, the positive judgement criteria for the IFA assay is over a titer of 80. If the Cut-off value is the same, the interpretation of the results of this experiment seems to change. Please clarify this point.

2.       The authors performed the FRNT50 assay on 10 cases that were strongly positive for IFA, so why did you not perform the FRNT50 assay on ZC03? ZC03 also shows a strong positivity (a titer of ZC03 is 60).

3.       Line 143-148. The authors compared seroprevalence in Vietnam specifically with that in Pakistan, but in the Pakistani study seroprevalence was examined by measuring neutralizing antibodies (Reference 10). Is it okay with a direct comparison with your studies that were investigated using the IFA assay? I think that it is better to perform the FRNT50 assay in all cases, not limited to the strongly IFA-positive cases, and compare them.

Minor comments

1.       Line 102. Please change “FRTN50” to “FRNT50”.

2.       Line 138-140. Authors should add the citing paper on the seroprevalence of each country to the references.

3.       Line 52, 140, 147, and 162-163. Reference No.11 is not related to the content of the text. I think the author is confusing Ref No.10 with No.11. Please correct it. In addition, please check carefully all references for numbering discrepancies.

4.       Line 156-157. It is a comment. The human-to-human transmission route of SFTSV is mainly believed to a contact to patients' blood or body fluids. Even if there are a small number of asymptomatic cases, I don't think they pose a threat enough to affect a SFTS outbreak control. Isn't that a bit of a leap of logic?

Author Response

Major comments

  1. About the section of Material and Methods. The authors should describe the cut-off level (value) for considering positive in the assay of IFA and ELISA. It is believed that the authors used the experimental system of Reference 4. According to Reference 4, the positive judgement criteria for the IFA assay is over a titer of 80. If the Cut-off value is the same, the interpretation of the results of this experiment seems to change. Please clarify this point.

: Dear reviewer,

Thank you very much for your commentary.

We studied the kinetics of serological response in patients with SFTS and published it at Viruses (reference 4. Ra, S.H.; Kim, M.J.; Kim, M.C.; Park, S.Y.; Park, S.Y.; Chung, Y.P.; et al. Kinetics of Serological Response in Patients with Severe Fever with Thrombocytopenia Syndrome. Viruses. 2021, 1 (1), 6).

Reference 4 was study about South Korea and the positive of the IFA assay is over a titer of 80 in this paper because South Korea is an endemic region of SFTSV and measured IgM and IgG in SFTS patients.

In this study, the positive IFA assay is not over a titer of 80 (not tightly judged) because Vietnam is not an endemic region of SFTSV and measured IgM and IgG in healthy residents.

However, we also measured IgM and IgG using ELISA for double check about positive samples by IFA assay 

“We used a 2-step approach to detect antibodies against SFTSV. 
First, we screened the samples for SFTSV IgG by using a IFA and did SFTSV ELISA kit (Bore Da Biotech Co., Ltd., http://www.boreda.com), which employs SFTSV nucleocapsid protein (NP) as the viral antigen”.
And our results showed that ELISAs for anti-SFTSV IgM or IgG were positive about 26 serum samples, were positive by IFA (IgM or IgG) and we think that 26 samples were positive of SFTSV although some samples are not over a titer of 80.

  1. The authors performed the FRNT50 assay on 10 cases that were strongly positive for IFA, so why did you not perform the FRNT50 assay on ZC03? ZC03 also shows a strong positivity (a titer of ZC03 is 60).

: Thank you very much for your commentary. Actually, we want to do an IFA about ZC03. But, unfortunately, we cannot do IFA  about ZC03 due to technical problems (the tube of ZC03 is broken and leak of the serum of ZC03).

  1. Line 143-148. The authors compared seroprevalence in Vietnam specifically with that in Pakistan, but in the Pakistani study seroprevalence was examined by measuring neutralizing antibodies (Reference 10). Is it okay with a direct comparison with your studies that were investigated using the IFA assay? I think that it is better to perform the FRNT50 assay in all cases, not limited to the strongly IFA-positive cases, and compare them.

: Thank you very much for your commentary and we fully agree with you.          IFA and ELISA methods usually used for seroprevalence and neutralizing antibodies assay used for confirmation of infection (reference 3, 11, 13, and 14).

Therefore, we used a 2-step (IFA and ELISA) approach to detect antibodies against SFTSV and confirmed SFTSV infection using neutralizing antibodies assay (FRNT50) in this study. 

“First, we screened the samples for SFTSV IgG by using a IFA and did SFTSV ELISA kit. and our results showed that the seroprevalence of anti-SFTSV IgM or IgG was observed to be 3.64% (26/714), and we did neutralizing antibodies assay for confirmation SFTSV infection about 26 positive samples and the titer of neutralizing antibodies against SFTSV ranged from 15.5 to 55.9 in Vietnam”.

Reference 11 (study of Pakistan) did ELISA and result of ELISA showed that Pakistan had a high seroprevalence (46.7%, 95% CI 44.3%-49.1%) of SFTSV, although Pakistan is not endemic region of SFTSV and SFTSV infection were confirmed using MNT, which revealed a low prevalence (2.5%, 95% CI 1.9%-3.4%).
Although we did not FRNT50 about whole samples (714 serum samples), if 26 samples, which were positive of IgM or IgG (26/714, 3.64%), have neutralizing antibodies, we think that 26 serum samples are positive of SFTSV.

So, we suggest that the seroprevalence rate in Vietnam is similar to that in Pakistan (line 35 and 36).

Minor comments

  1. Line 102. Please change “FRTN50” to “FRNT50”.

: Thank you very much for your correction. We correct “FRNT50” in line 102. 

  1. Line 138-140. Authors should add the citing paper on the seroprevalence of each country to the references.

: Thank you very much for your commentary. Below are contents in reference 11. 
“In ELISA-based estimates of SFTSV in the human population reported from different areas of East Asian countries, seroprevalence has ranged from 0.23% to 9.17% in China (7), from 1.9% to 7.7% in Korea (8, 9), and from 0.14% to 0.3% in Japan (10, 11)”. 

So, we put reference 11 in line 138-140. However, reviewers want to add the citing paper on the seroprevalence of each country, we will put these in line 138-140 (China (7), Korea (8, 9), and Japan (10, 11)).

  1. Line 52, 140, 147, and 162-163. Reference No.11 is not related to the content of the text. I think the author is confusing Ref No.10 with No.11. Please correct it. In addition, please check carefully all references for numbering discrepancies.

: Thank you very much for your commentary. I am sorry and confused by Ref. No. 10 with. No. 11 and correct it and we also recheck all reference in manuscript and correct.

  1. Line 156-157. It is a comment. The human-to-human transmission route of SFTSV is mainly believed to a contact to patients' blood or body fluids. Even if there are a small number of asymptomatic cases, I don't think they pose a threat enough to affect a SFTS outbreak control. Isn't that a bit of a leap of logic?

: Thank you very much for your commentary. Reference 3 is about “seroprevalence of antibodies specific for severe fever with thrombocytopenia syndrome virus and the discovery of asymptomatic infections in Henan

Province, China” and authors of reference 3 concluded below.

Conclusions This study identified a relatively high incidence of SFTSV-specific antibody seropositivity in healthy people in Xinyang city. Moreover, our data provide the first evidence for asymptomatic SFTSV infections, which may have significant implications for SFTS outbreak control”.

We also agree with these authors. So, we also conclude that with these authors.

Please find a revised manuscript. 

Round 2

Reviewer 1 Report (Previous Reviewer 2)

Thank the authors to the answers.

I think that introduction could be implemented wiith the information of other studies that analyzed this problem.There is a systematic review of this topic with interesting information. Here the reference " Liu, S., Chai, C., Wang, C., Amer, S., Lv, H., He, H., ... & Lin, J. (2014). Systematic review of severe fever with thrombocytopenia syndrome: virology, epidemiology, and clinical characteristics. Reviews in medical virology, 24(2), 90-102."In the introduction I suggest to provide some information of the virus'features "Yu XJ, Liang MF, Zhang SY, et al. Fever with thrombocytopenia associated with a novel Bunyavirus in China. New England Journal of Medicine 2011; 364: 15231532.., "Xiong WY, Feng ZJ, Matsui T, et al. Risk assessment of human infection with a novel Bunyavirus in China. Western Pacific Surveill and Response Journal 2012; 3: 6974".

Author Response

I think that introduction could be implemented wiith the information of other studies that analyzed this problem. 

There is a systematic review of this topic with interesting information. 

Here the reference

  1. Yu, X.J.; Liang, M.F.; Zhang, S.Y.; Liu, Y.; Li, J.D.; Sun, Y.L.; Zhang, L.; Zhang, Q.F.; Popov, V.L.; Li, C.; et al. Fever with thrombocytopenia associated with a novel bunyavirus in China. N. Engl. J. Med. 2011, 364, 1523–1532.
  2. Liu, S.; Chai, C.; Wang, C..; Amer, S., Lv, H.; He, H; Sun, J.; Lin, J. Systematic review of severe fever with thrombocytopenia syndrome: virology, epidemiology, and clinical characteristics. Reviews in medical virology2014, 24(2), 90-102. Rev Med Virol. 2014, 24(2):90-102.

: Thank you very much for your advice about references. We put these (reference 1 and 5).

"In the introduction I suggest to provide some information of the virus'features

  1. Yu XJ, Liang MF, Zhang SY, et al. Fever with thrombocytopenia associated with a novel Bunyavirusin China. New England Journal of Medicine 2011; 364: 1523-1532.,
  2. Xiong, W.Y.; Feng, Z.J.; Matsui,T.; Foxwell, A.R. Risk assessment of human infection with a novel bunyavirus in China. Western Pac Surveill Response J. 2012; 3(4):61-66.

: Thank you very much for your commentary. We put these (reference 1 and 4) and also put below in line 43 and 45. 

“Severe fever with thrombocytopenia syndrome virus (SFTSV) is an enveloped and tri-segmented (large [L], middle [M], and small [S]) negative-strand RNA virus and was first reported in rural areas of Hubei and Henan provinces in central China in 2009 [1, 4]”.

Reviewer 2 Report (Previous Reviewer 1)

Comments

1.      In the response to Major comment No. 3, the authors state that they tested neutralizing antibodies by FRNT50 assay in 26 cases that were positive by IFA and/or ELISA; however, the authors examined only 10 cases. If the FRTN50 assay is performed to confirm SFTSV infection, shouldn't the authors perform the FRTN assay on all 26 samples?

2. Line 138-139. Please add the citing paper on the seroprevalence of each country. 

Author Response

Comments
1.  In the response to Major comment No. 3, the authors state that they tested neutralizing antibodies by FRNT50 assay in 26 cases that were positive by IFA and/or ELISA; however, the authors examined only 10 cases. 
If the FRTN50 assay is performed to confirm SFTSV infection, shouldn't the authors perform the FRTN assay on all 26 samples?
: Thank you very much for your commentary. Yes! I fully agree with you and sorry about not performance FRNT50 about 26 samples. 

We should perform the FRNT50 on all 26 samples. But, we only performed FRNT50 on 10 samples.                                                                                             The reasons are below (2). 
First, SFTSV and SARS-CoV-2 are level 3 pathogens, and we cannot room for performance on all 26 samples at that time because SARS-CoV-2 is big problem at that time and most of efforts and room are concentrating about study of SARS-CoV-2. Therefore, we can do 10 samples. 
Second, our previous study (Reference 20. Yoo JR, Kim JY, Heo ST, Kim J, Park HJ, Lee JY, et al. Neutralizing Antibodies to Severe Fever With Thrombocytopenia Syndrome Virus Among Survivors, Non-Survivors and Healthy Residents in South Korea. Front Cell Infect Microbiol. 2021;11:649570), we found that the titers of neutralizing antibodies in healthy residents who were positive for SFTSV IgG. 
So, we think that 26 samples may have the titers of neutralizing antibodies. 
But, sorry again about not performance all 26 samples. 

2. Line 138-139. Please add the citing paper on the seroprevalence of each country. : Thank you very much for your commentary about references.             We put these (reference 15 to 19) in manuscript. 
15. Li,  P.; Tong,  Z.D.; Li, K.F.; Tang,  A.; Dai,  Y.X.; Yan,  J.B. Seroprevalence of severe fever with thrombocytopenia syndrome virus in China: A systematic review and meta-analysis. PLoS One 2017, 12:e0175592.  
16.Han,  M.A.; Kim,  C.M.; Kim,  D.M.; Yun,  N.R.; Park,  S.W.; Han,  M.G.; et al. 
Seroprevalence of severe fever with thrombocytopenia syndrome virus antibodies in rural areas, South Korea. Emerg Infect Dis. 2018, 24(5): 872-874. 
17. Kim,  K.H.; Ko,  M.K.; Kim,  N.; Kim,  H.H.; Yi.  J. Seroprevalence of severe fever with thrombocytopenia syndrome in southeastern Korea, 2015. J Korean Med Sci. 2017, 32:29-32.  
18. Kimura,  T.; Fukuma,  A.; Shimojima,  M.; Yamashita,  Y.; Mizota,  F.; Yamashita, M.; et al. Seroprevalence of severe fever with thrombocytopenia syndrome (SFTS) virus antibodies in humans and animals in Ehime prefecture, Japan, an endemic region of SFTS. J Infect Chemother. 2018,24:802-806.  
19. Gokuden,  M.; Fukushi,  S.; Saijo,  M.; Nakadouzono,  F.; Iwamoto,  Y.; Yamamoto, M.; et al. Low seroprevalence of severe fever with thrombocytopenia syndrome virus antibodies in individuals living in an endemic area in Japan. Jpn J Infect Dis. 2018,71:225-228.

This manuscript is a resubmission of an earlier submission. The following is a list of the peer review reports and author responses from that submission.

Round 1

Reviewer 1 Report

To the authors,

The authors report measuring the prevalence of anti-SFTSV antibodies in healthy individuals to estimate the disease burden of SFTS in Vietnam. This is a useful report because such an epidemiological study in Vietnam has not been reported before.

Major comments

  1. Table 1 and Fig. 1 show the IFA images and results, but it is difficult to understand that (1) ZC397 was determined to be positive and (2) ZC613 and ZC674 were determined to be negative in the IgM assay. It appears that the image findings are not consistent with the results. This may lead to misunderstanding by the reader, and the interpretation should be corrected or comments should be added on this point.
  2. Page 5, line 131. ZH129 must be a mistake for ZH128. Please check it.
  3. I have not found any reports of SFTS cases in Vietnam other than the author's (ref. No 6). The authors have found in this study that the seroprevalence of SFTSV infection in Vietnam is similar to that in China and South Korea: however, why are there so few reports of SFTS patients in Vietnam? Authors should add comments on this conflicting point.
  4. The authors mention the large disease burden of SFTS in Vietnam in terms of seroprevalence. However, if the incidence of SFTS patients in Vietnam is low, it cannot be said that the disease burden of SFTS in Vietnam is large. If you have any data showing that the number of SFTS patients is increasing in Vietnam, please indicate.

Minor comments

  1. Table 1. Please align the lines of ZH048.

Author Response

Major comments
1. Table 1 and Fig. 1 show the IFA images and results, but it is difficult to understand that (1) ZC397 was determined to be positive and (2) ZC613 and ZC674 were determined to be negative in the IgM assay. It appears that the image findings are not consistent with the results. 
This may lead to misunderstanding by the reader, and the interpretation should be corrected or comments should be added on this point.

: First of all, thank you very much for your valuable commentary. 
Two professors (Prof. Kee and Prof. Lee) readied IFA for cross check. 
Dr. Kee readied +/- (positive and negative) on IgM about ZC397 and Prof. Lee readied positive about this sample and finally decided positive because Dr. Lee did ELISA about ZC397 and the result of ELISA was positive.

If you think this is not positive, we can change from positive to negative about ZC397.

For ZC613, ELISA data of ZC613 was negative. So, we determined negative.
If you think this was positive, we can change “negative” to “positive” about ZC613.

For ZC674, ELISA data of ZC674 was positive. So, we corrected from “negative” to “positive”.

Thank you again for your valuable commentary. 

2. Page 5, line 131. ZH129 must be a mistake for ZH128. Please check it.

: Thank you very much for your careful check about this (ZH 129 and ZH128).
I'm really sorry. This is my mistake and I corrected from “ZH129” to “ZH128”. 

3. I have not found any reports of SFTS cases in Vietnam other than the author's (ref. No 6). 
The authors have found in this study that the seroprevalence of SFTSV infection in Vietnam is similarto that in China and South Korea: however, why are there so few reports of SFTS patients in Vietnam? Authors should add comments on this conflicting point.

: I am fully in agreement about the commentary of the reviewer. 
Yes! Ref. No. 6 is the first reported SFTS patient in Vietnam and I think that more cases could be found in Vietnam because this is recently reported in Vietnam because after first reported SFTS patients in Vietnam, SFTS patients in other Southeast Countries such as Thailand, which is a neighboring country, are reported in 2020 [1-3] and A. testudinarium, a vector of SFTSV, has been found in Vietnam [4].
[1. Severe Fever with Thrombocytopenia Syndrome Virus: The First Case Report in Thailand, The Bangkok Medical Journal Vol. 16, No. 2; September 2020 “We   report   a   case   of   a   70-year-old   Thai   woman   with   severe   fever   and thrombocytopenia syndrome, who had lost all seven of her cats from sickness over a week. Diagnosis was established by the detection of viral RNA in serum via real-time polymerase chain reaction. Her symptoms improved after taking doxycycline orally and supportive treatment”

2. Genotypic Heterogeneity of Orientia tsutsugamushi in Scrub Typhus Patients and Thrombocytopenia Syndrome Co-infection, Myanmar. Emerg. Infect. Dis. 2020, 26 (8), 1878-1881.).

3. Human Case of Severe Fever with Thrombocytopenia Syndrome Virus Infection, Taiwan 2019. Emerg. Infect. Dis. 2020, 26 (7), 1612-1614.

4. Endemic severe fever with thrombocytopenia syndrome, Vietnam. Emerg. Infect. Dis. 2019, 25 (5) :1029-1031.]

Ref. No. 9 is serology study of SFTSV in Pakistan.
This study confirmed SFTSV infection using MNT (microneutralization test), which revealed 2.5% prevalence (human serum samples size is 1,657) in Pakistan and suggested the potential risk for infection from SFTSV and SFTSV-related viruses in Pakistan

[Ref. Serologic Evidence of Severe Fever with Thrombocytopenia Syndrome Virus and Related Viruses in Pakistan. Emerg. Infect. Dis. 2020, 26 (7),1513-1516].

The seroprevalence of anti-SFTSV IgM or IgG was observed to be 3.64% (26/714) in Vietnam and we think that prevalence rate is similar with Pakistan.

Therefore, we suggest that the disease burden is similar to that in China (0.23% to 9.17%), South Korea (1.9% to 7.7%), and Pakistan (2.5%). 

However, If you do not agree, we can change this from “the findings suggest that the disease burden in Vietnam is similar to that in China, South Korea, and Pakistan.” to “the findings suggest that the disease burden in Vietnam could be increased in the future”.

4. The authors mention the large disease burden of SFTS in Vietnam in terms of seroprevalence. However, if the incidence of SFTS patients in Vietnam is low, it cannot be said that the disease burden of SFTS in Vietnam is large. If you have any data showing that the number of SFTS patients is increasing in Vietnam, please indicate.

: SFTS patients in Southeast Asia were reported from 2018 and confirmed SFTSV infection in Pakistan is 2.5% using MNT and the seroprevalence of anti-SFTSV IgM or IgG was observed to be 3.64% (26/714) in our study. 

So, we think that the number of SFTS patients could be increased in this area in the future based on seroprevalence study of Vietnam, Pakistan, and Thailand and suggesting that the disease burden is similar to  that in China (0.23% to 9.17%), South Korea (1.9% to 7.7%), and Pakistan (2.5%). 

However, If you do not agree, we can change this from “the findings suggest that the disease burden in Vietnam is similar to that in China, South Korea, and Pakistan.” to “the findings suggest that the disease burden in Vietnam could be increased in the future”.

Minor comments
1. Table 1. Please align the lines of ZH048.
: Thank you very much for your commentary and I do align the line of ZH048. 

Reviewer 2 Report

Thank you for the opportunity to revise this manuscript.

Manuscript presentes some limititation and it needs a thorough revision. I do not consider it suitable for publication in its current form.

Here some major concerns.

The introduction should be expanded.  It’s important to deepen the epidemiology of the disease. Since SFTSV was first identified, epidemics have occurred in several East Asian countries. With the escalating incidence of SFTS and the rapid, worldwide spread of SFTSV vector, it is clear this virus has pandemic potential and presents an impending global public health threat.

Being a public health problem, it is useful to propose some solutions. The impact of the covid 19 pandemic has rekindeled the debate regarding the use of substances with antiviral properties. Severe fever with thrombocytopenia syndrome (SFTS) is an emerging infectious zoonosis in China and other countries in southeast Asia so, to stem the spread of new viral infections, mentioning the use of substances with antiviral propierts could be a useful food for thought. A good recent version in support are “Sinopoli, A., Isonne, C., Santoro, M. M., & Baccolini, V. (2022). The effects of orally administered lactoferrin in the prevention and management of viral infections: A systematic review. Reviews in medical virology, 32(1), e2261”, “Behboudi, E., Zeynali, P., Zahedian Nezhad, N., & Hamidi Sofiani, V. (2022). Vitamin A and Viral Infection in Critical Care. Jorjani Biomedicine Journal, 10(1), 67-92.”.

Methods should be thoroughly investigated. Information on the study population is lacking: age, sex, eventuall comorbidity ( it is not enough to report the data in the table).

I therefore recommend that we review the discussion in light of this.

Author Response

Manuscript presentes some limititation and it needs a thorough revision. I do not consider it suitablefor publication in its current form.

Here some major concerns.

The introduction should be expanded.  
It’s important to deepen the epidemiology of the disease. Since SFTSV was first identified, epidemics have occurred in several East Asian countries. With the escalating incidence of SFTS and the rapid, worldwide spread of SFTSV vector, it is clear this virus has pandemic potential and presents an impending global public health threat.
Being a public health problem, it is useful to propose some solutions. 
The impact of the covid 19 pandemic has rekindeled the debate regarding the use of substances 
with antiviral properties. 
Severe fever with thrombocytopenia syndrome (SFTS) is an emerging infectious zoonosis in China and other countries in southeast Asia so, to stem the spread of new viral infections, mentioning the use of substances with antiviral propierts could be a useful food for thought. 
A good recent version in support are “Sinopoli, A., Isonne, C., Santoro, M. M., & Baccolini, V. (2022).The effects of orally administered lactoferrin in the prevention and management of viral infections: A systematic review. Reviews in medical virology, 32(1), e2261”, “Behboudi, E., Zeynali, P., Zahedian Nezhad, N., & Hamidi Sofiani, V. (2022). Vitamin A and Viral Infection in Critical Care. Jorjani Biomedicine Journal, 10(1), 67-92.”.

: Thank you very much for your commentary. 
We expand more about the introduction between 46 and 53 in manuscript according to your commentary. 
“Severe fever with thrombocytopenia syndrome virus (SFTSV) is a tickborne virus of the genus Phlebovirus and family Phenuiviridae [1-3]. 
SFTS is characterized by acute high fever, thrombocytopenia, leukopenia, elevated serum hepatic enzyme levels, gastrointestinal symptoms, and multiorgan failure, has a 16.2 to 30% mortality rate, and effective antiviral therapy for SFTS virus (SFTSV) has not been available [1-4]. 
Atypical signs and symptoms as well as asymptomatic SFTSV infections have also been identified in patients and healthy people [2, 3].”

For antiviral therapy for SFTS virus (SFTSV), effective antiviral therapy for SFTSV has not been available.
[1. Successful treatment of rapidly progressing severe fever with thrombocytopenia syndrome with neurological complications using intravenous immunoglobulin and corticosteroid. Antivir. Ther. 2016, 21, 637–640. 
2. Effect of early plasma exchange on survival in patients with severe fever with thrombocytopenia syndrome: A multicenter study. Yonsei Med. J. 2017, 58, 867–871. 
3. Park, S.Y.; Choi, W.; Chong, Y.P.; Park, S.W.; Wang, E.B.; Lee, W.J.; Jee, Y.; Kwon, S.W.; Kim, S.H. Use of plasma therapy for severe fever with thrombocytopenia syndrome encephalopathy. Emerg. Infect. Dis. 2016, 22, 1306–1308.]

So, we would like to put suggestion references from you after accumulation of more data in antiviral therapy for SFTS virus. 

Methods should be thoroughly investigated. 
Information on the study population is lacking: age, sex, eventuall comorbidity (it is not enough to report the data in the table).

: Thank you very much for your commentary. In this study, we investigated the seroprevalence of SFTSV in 714 healthy residents in Thua Thien Hue and Quang Nam Province, Vietnam, encompassing areas of large forests, hilly landscapes, and grasslands, from October 1, 2017, to September 30, 2018. Most of the 714 residents were farmers, with a tick exposure risk. 

So, we have no comorbidity and laboratory data about 714 healthy residents because these are not SFTS patients. 

Round 2

Reviewer 1 Report

To the authors,

<Comments>

About Fig 1 and Table 1 (IFA method). The authors replied that their IFA results (ZC397, ZC613, and ZC674) could be changed to either positive or negative to suit my comments. Judgment of the result should be made according to a strictly defined method, and the author's way of thinking is not scientific at all.

In addition, IFA and ELISA are completely independent experimental systems. It is completely incorrect to change the judgment of the IFA result by referring to the ELISA result.

The credibility of this study has been greatly impaired, and it cannot be said to be a scientific paper.

Reviewer 2 Report

"So, we would like to put suggestion references from you after accumulation of more data in antiviral therapy for SFTS virus".
I really don't understand the meaning of this sentence. I suggested underlining the debate regarding the use of substances with antiviral properties that were rekindled during the recent pandemic covid 19.
Being a public health problem I think it's only right for the researchers to provide a possible solution. And I suggested highlighting the possible use of substances with antiviral properties against this tickborne virus of the genus Phlebovirus according to the references that  I've already hinted.
In addition, the epidemiology of the disease is not yet sufficiently explored.
In the methods you have to report some news about the population: mean age, % of occupation, etc...